# Multi-fineness Boundaries and the Shifted Ensemble-aware Encoding for Point Cloud Semantic Segmentation

## ABSTRACT

Point cloud segmentation forms the foundation of 3D scene understanding. Boundaries, the intersections of regions, are prone to mis-segmentation. Current point cloud segmentation models exhibit unsatisfactory performance on boundaries. There is limited focus on explicitly addressing semantic segmentation of point cloud boundaries. We introduce a method called Multi-fineness Boundary Constraint (**MBC**) to tackle this challenge. By querying boundaries at various degrees of fineness and imposing feature constraints within these boundary areas, we enhance the discrimination between boundaries and non-boundaries, improving point cloud boundary segmentation. However, solely emphasizing boundaries may compromise the segmentation accuracy in broader non-boundary regions. To mitigate this, we introduce a new concept of point cloud space termed **ensemble** and a Shifted Ensemble-aware Perception (**SEP**) module. This module establishes information interactions between points with minimal computational cost, effectively capturing direct point-to-point long-range correlations within ensembles. It enhances segmentation performance for both boundaries and non-boundaries. We conduct experiments on multiple benchmarks. The experimental results demonstrate that our method achieves performance surpassing or comparable to state-of-the-art methods, validating the effectiveness and superiority of our approach.

## CCS CONCEPTS

• **Computing methodologies → Scene understanding**.

## KEYWORDS

3d boundary query, feature constraint, long-range correlations, 3d segmentation, point clouds, scene understanding

## 1 INTRODUCTION

Point cloud semantic segmentation is a fundamental task in understanding 3D scenes. Due to point clouds' unordered and unstructured nature, mature convolutional methods [19, 20] cannot be directly applied to them. [9, 33, 61] preprocess point clouds into structured voxels suitable for 3D convolutions. [7, 12, 15] project point clouds from different views and extracts features using 2D convolutions. PointNet [35] introduces the symmetric functions,

**Unpublished working draft. Not for distribution.**

enabling direct input of raw point clouds into neural networks. Subsequently, numerous excellent methods [32, 36, 38] have emerged. However, most of them are unsatisfactory for segmenting boundaries [43]. Boundaries are transitional areas between regions, blending features from points belonging to various categories. These mixed features are characterized by complexity and ambiguity. As the network deepens, these ambiguous features encompassing information from multiple semantic categories inevitably spread within boundary regions, resulting in an unsatisfactory performance on semantic segmentation at boundaries [10].

Boundary plays a critical role in semantic segmentation. However, explicit research on semantic segmentation of point cloud boundaries remains relatively limited compared to 2D image segmentation. JSENet [23] and BGE [10] build a boundary prediction branch network to assist segmentation. CBL [43] utilizes contrastive learning to optimize the representation of boundary features. The commonality among these methods is the requirement to pre-know the set of boundary points by querying. The quality of the boundary query will affect the performance of boundary semantic segmentation. In 2D images, pixels are semantically connected and continuous, and a set of pixels can accurately define boundaries. However, the increased dimensions and the sparsity of point clouds make objects have more complex spatial interactions with others, bringing challenges to the boundary definition. [10, 43] use the neighbor query method KNN (K-Nearest Neighbors) to iterate over points and label the point as a boundary point if there are points with different classes in its neighbor. The accuracy of boundaries heavily depends on the single neighbor query setting, which have a non-negligible impact on subsequent semantic segmentation.

Different query degrees of fineness result in different boundaries. High-confidence boundaries can be obtained with a high degree of fineness query (i.e., reducing the neighbor query range or the number of query points), but some boundary points may be missed. Conversely, with a low degree of fineness query, incorrect boundary delineation may occur, but the integrity of boundary points increases. Given this, we propose a method called Multi-fineness Boundary Constraint (**MBC**). As shown in the bottom-right of Figure 1, it conducts boundary queries at different degrees of fineness to obtain more precise and complete boundaries. Boundary points with the same category and located nearby should exhibit similar features. MBC randomly selects multiple points as key points for feature constraints in the boundary neighbors. It encourages similarity among points with the same class and dissimilarity between points with different classes, enhancing the discrimination between boundaries and non-boundaries. Without additional networks, this approach reduces the negative impact of erroneous boundary queries and improves the segmentation accuracy of boundaries.

Introducing additional boundary feature constraints may divert the model's attention from semantic segmentation, affecting the segmentation performance of extensive non-boundary areas [43].

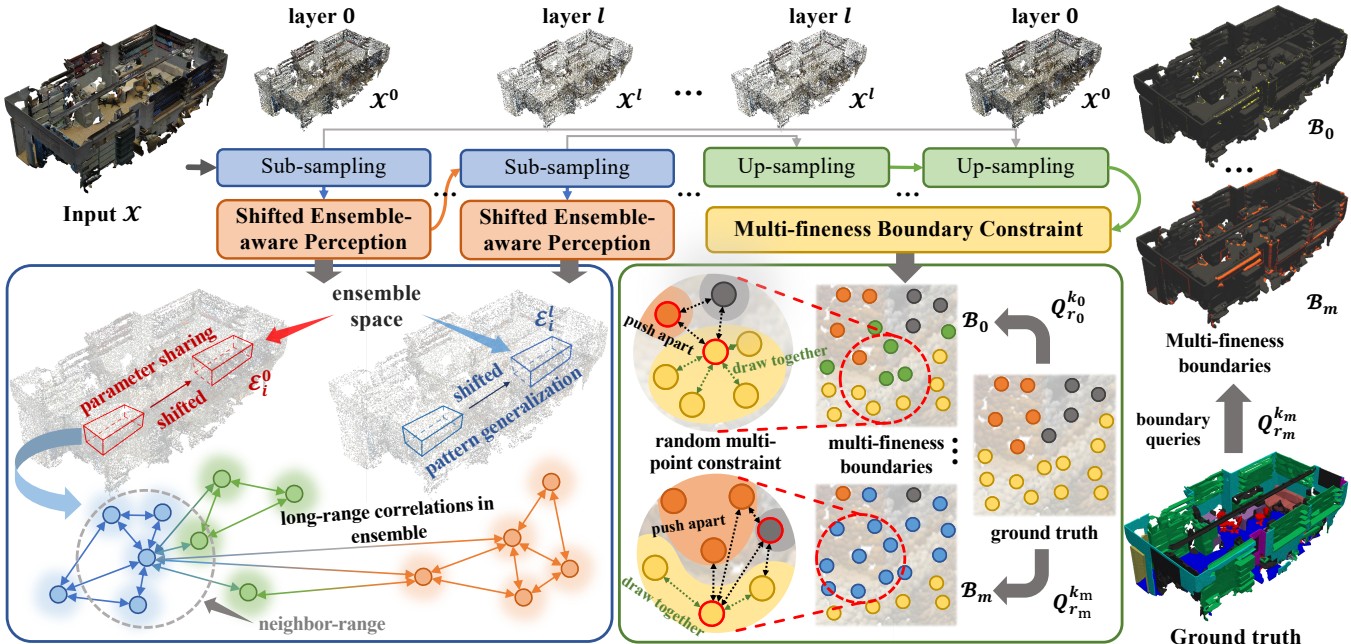

**Figure 1: The overall architecture of MBSE. Bottom-right: the schematic diagram of the MBC. The green and blue points represent boundary points queried at different degrees of fineness. Points with red contours indicate constraint key points. Bottom-left: the schematic diagram of the SEP. Ensembles are randomly sampled on the whole input point cloud, not requiring additional operations such as voxelization. For clarity, cuboids depict ensembles, and some interaction arrows are omitted here.**

Balancing global segmentation accuracy is another challenge we encounter. Long-range correlations refer to dependencies between distant points, such as geometric and semantic relationships [25]. They aid the model in comprehending the semantic structure and context of the entire point cloud, enabling a broader perception of the distinctions between boundaries and non-boundaries, thereby enhancing overall segmentation performance. Most models extract and aggregate features within neighbors consisting of several tens of points. They rely on the deepening of the network to expand the receptive field progressively. This indirect way makes capturing long-range correlations among points difficult, potentially resulting in the blurring or even loss of crucial features during multiple abstraction processes. As illustrated in the bottom-left of Figure 1, we introduce a novel concept in point cloud space called **ensemble**, along with a shifted ensemble-aware perception module (**SEP**) to address this issue. Unlike methods like BGE [10] that employ masks to prevent feature aggregation, we encourage information exchange between points. The point cloud is randomly partitioned into multiple ensembles with an equal point count. SEP captures long-range correlations within each ensemble, enabling feature abstraction and aggregation beyond local neighbors. For better global features, SEP generalizes the ensemble feature pattern to the global point cloud space with minimal computational cost through shifting and parameter-sharing on multi-scale point clouds, which improves the overall semantic segmentation performance.

The MBC and the SEP are detailed in Section 3.1 and Section 3.2, respectively. We name our method **MBSE** (Multi-fineness Boundaries and the Shifted Ensemble-aware Encoding framework), which

enhances the segmentation performance of both boundaries and non-boundaries. Our main contributions are as follows:

- We propose a method called multi-fineness boundary constraint (**MBC**). Without additional networks, it enhances the feature discrimination between boundaries and non-boundaries, assisting the discriminator in making better segmentation decisions for boundaries.
- We propose a new concept in point cloud space called **ensemble** and a shifted ensemble-aware perception (**SEP**) module. SEP delicately captures direct point-to-point long-range correlations within ensembles. Through shifting and parameter sharing, SEP generalizes the ensemble-level patterns to the entire point cloud with low computational cost, enhancing the overall semantic segmentation accuracy.
- The extensive experimental results demonstrate that **MBSE** notably improves the performance of baselines, achieving state-of-the-art or highly competitive performance on multiple benchmarks with an increase of just 1M parameters and 0.5 GFLOPs in computational cost.

## 2 RELATED WORK

### 2.1 Boundary Segmentation

Research on boundary semantic segmentation of 2D images has a long history. Initially, researchers use low-level image features such as color to predict boundaries. SBD [17] introduces the first semantic boundary dataset for 2D images, accelerating the development

of 2D boundary segmentation. HFL [2] establishes a two-stage prediction network for separately predicting boundaries and boundary categories. CaseNet [57] proposes an end-to-end deep semantic edge learning architecture based on ResNet [19] with category awareness. PED [22] aggregates semantic and instance boundary segmentation tasks into a multi-branch network. SEAL [58] proposes a method of correcting noisy labels to help networks generate high-quality boundaries.

Compared to the mature 2D image boundary segmentation, relatively fewer studies explicitly focus on point cloud boundary segmentation. Some methods [10, 23] divide 3D semantic segmentation into boundary prediction and semantic segmentation subtasks. The segmentation accuracy of these methods is closely related to the quality of boundary prediction, incorrect boundary predictions can adversely affect semantic segmentation. CBL [43] demonstrates the poor performance of current point cloud semantic segmentation models in boundary segmentation and proposes a method based on contrastive learning to enhance boundary feature representation. These explicit boundary semantic segmentation methods rely on querying to obtain boundary points. However, none of the above methods realize the impact of boundary queries on semantic segmentation. Our work does not treat the boundary prediction as a subtask. We introduce feature constraints on boundaries at different degrees of fineness to optimize the feature representation and enhance the semantic discrimination between boundaries and non-boundaries, aiding the discriminator in making better segmentation decisions on boundary areas.

## 2.2 Point Cloud Feature Abstraction

The inherent non-structural and unordered characteristics of point clouds pose challenges for feature abstraction. Various approaches have been developed for preprocessing point clouds to structured data. Voxel-based methods [31, 47, 54, 61] transform point clouds into fixed-size voxels, treating them as the smallest processing units for feature abstraction. SEGCloud [44] utilizes a 3D fully convolutional neural network (CNN) to extract features from point cloud voxels and then restores the prediction results to the original point cloud resolution through trilinear interpolation. To mitigate unnecessary computations caused by sparse point clouds when applying 3D dense convolution, OctNet [41] uses an unbalanced octree to partition the point cloud hierarchically, with its leaf nodes storing feature representations extracted by 3D convolution. Voxel-based methods can directly utilize mature CNNs to extract point cloud features, but the voxelization process may lead to the loss of information such as spatial position.

Methods based on multi-view projection [7, 15, 16, 42, 49] project 3D point clouds into 2D images from different viewpoints. CNNs are employed to extract and fuse features that can be utilized for downstream tasks. SnapNet [3] generates RGB and depth images, employs fully convolutional networks for 2D image feature abstraction and labeling, and then reprojects them back to the point cloud. SnapNet-R [11] improves upon SnapNet by independently using FuseNet [18] to extract features and performing semantic segmentation on multi-view images. Methods based on multi-view are susceptible to the projection angles. Additionally, this method may lose features due to internal occlusions within point clouds.

Point-based methods [24, 48, 55, 59] directly extract features from the raw point cloud, making them a current research hotspot in point cloud semantic segmentation. PointNet++ [36] introduces hierarchical and multi-scale grouping structures to capture features at multiple scales and local neighbors. Subsequently, Point Transformer [60] and StratifiedTransformer [25] establish transformer-based frameworks for point cloud semantic segmentation, which achieve excellent performances. Meanwhile, MLP-based methods PointNext [38] and PointVector [6] demonstrate the competitiveness of MLPs (Multi-Layer Perceptions) and Transformer [46]. However, the methods mentioned above progressively aggregate local features in neighbors, making it difficult for distant points to interact. Although Transformer structures have the advantage of large receptive fields, they still apply self-attention at a local feature level due to the high volume of point cloud data. The locally aggregated features may blur the information of key points. Unlike existing works, we focus on the direct interaction between points. We generalize the patterns to the entire point cloud through shifting and parameter-sharing with a low computational cost, capturing better local and global feature representations.

## 3 METHOD

In this section, we present our MBSE framework, comprising the multi-fineness boundary constraint (MBC) and the shifted ensemble-aware perception (SEP). The two components are described in Section 3.1 and 3.2, respectively.

## 3.1 Multi-fineness Boundary Constraint

Explicitly improving the performance of point cloud boundary segmentation requires accurate localization of boundaries. We observe that point cloud boundary queries can impact semantic segmentation performance, a problem that has yet to be systematically explored in previous research. Incorrect boundary queries can adversely affect boundary feature representations, thereby misleading semantic segmentation discriminators. In this section, we present a multi-fineness boundary constraint method that imposes feature similarity constraints on multiple sets of boundaries queried with different degrees of fineness. This approach enhances the semantic discrimination between boundaries and non-boundaries, improving the segmentation performance on boundaries.

### 3.1.1 Multi-fineness Boundary Query. [10, 43] uses KNN to query the point's neighbor and determines whether it is a boundary point according to the categories of the points in its neighbor, which has a severe drawback. Due to the uneven distribution of point clouds, KNN, which lacks distance restrictions, may group distant points into the same neighbor, leading to errors in boundary neighbor queries in sparse areas. We refer to this situation as the enclave phenomenon. To address this issue, we use a ball query to limit the query distance of the neighbors. Specifically, to classify a point $x_i$, which belongs to the point cloud $\mathcal{X}$, we query the $k$ nearest points of $x_i$ within the preset 3D spatial range. If there is a point that is inconsistent with the class of $x_i$, $x_i$ is identified as a boundary point. The set of boundary points $\mathcal{B}$ can be represented by the following formula:

$$\mathcal{B} = \{x_i \in \mathcal{X} \mid \exists x_j \in Q_r^k(x_i) \land l_i \neq l_j\}. \tag{1}$$

Here, $l_i$ represents the ground truth of $x_i$, and $Q_r^k(x_i)$ denotes the operation of querying the $k$ nearest points within a distance $r$ from $x_i$. Although the method alleviates the enclave phenomenon, inappropriate settings of $k$ and $r$ may lead to inaccurate boundary queries for point clouds. A high degree of fineness query (i.e., reducing $r$ or $k$) can enhance the precision of boundary queries but may overlook some boundary points. Conversely, a low degree of fineness query (i.e., increasing $r$ or $k$) can obtain more complete boundaries but might mistakenly classify non-boundary points as boundary points. Considering this, we conduct boundary queries at multiple degrees of fineness, obtaining multiple sets of boundaries by setting different values of $k$ and $r$. The multi-fineness boundary set $\mathcal{B}_m$ can be formalized as:

$$\mathcal{B}_m = \{x_i \in \mathcal{X} \mid \exists x_j \in Q_{r_m}^{k_m}(x_i) \wedge l_i \neq l_j\}, m = \{0, ..., M-1\}. \quad (2)$$

Where $M$ represents the number of fineness degrees. Conducting boundary queries at multiple degrees of fineness helps mitigate potential issues from a single query, such as boundary misclassification and incompleteness.

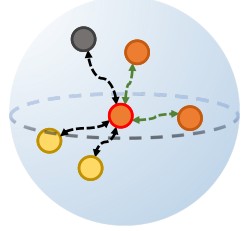 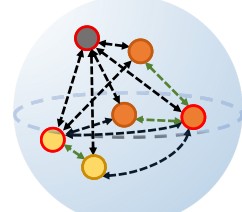

(a) Single center point      (b) Random multiple points

**Figure 2: Comparison between two constraints. The different colors of the points represent their respective classes. The points with red contours indicate constraint key points. The black and green dashed lines indicate the features being pushed away and drawn together, respectively.**

*3.1.2 **Random Multi-point Boundary Feature Constraints.***
Features serve as the basis for segmentation, and clear and distinguishable features are conducive to the segmentation discriminator making correct decisions. However, boundaries represent intersections of two or more objects with different categories. As local features are continuously extracted and aggregated, boundary regions experience feature fusion among points with different categories, leading to ambiguity in the region's features. This ambiguity poses a challenge for the point cloud semantic segmentation discriminator in accurately segmenting boundary points, which is one of the main reasons for the poor performance of current semantic segmentation models in boundary regions.

To address this issue, we impose feature constraints within the boundary neighbor whose center point is the boundary point. In the neighbor, we enhance the feature similarity between the constraint key points and the points belonging to the same category while reducing it otherwise. CBL [43] takes the center points as constraint key points. As shown in Figure 2a, The single center point constraint only imposes constraints between the constraint

key point and its neighboring points, neglecting the establishment of constraint relationships among other points except for the constraint key point. The similarity information of features within the boundary neighbor is underutilized, limiting the effectiveness of the feature constraints. As illustrated in Figure 2b, we extend the feature constraints to the entire boundary neighbor. Specifically, in a boundary neighbor, $q$ non-repeating points are randomly selected as constraint key points for $q$ times of feature constraints. Inspired by [8, 14, 34], the feature constraint for the boundary neighbor centered at $x$ ($x \in \mathcal{B}$) can be formulated as follows:

$$\mathcal{L}_x = -\frac{1}{q} \sum_{j \in R\left(Q_r^k(x)\right)} log \frac{\sum_{p \in Q_r^k(x) \wedge l_p = l_j} exp\left(-d(f_p, f_j)/\tau\right)}{\sum_{v \in Q_r^k(x)} exp\left(-d(f_v, f_j)/\tau\right)}. \quad (3)$$

Where $d$ represents the Euclidean metric. $\tau$ represents the constraint temperature. $R(\cdot)$ represents the operation of randomly non-repeated selection. Note that in low-density regions, there may be cases where the number of points is less than $k$ in some neighbor queries. In our implementation, the points are randomly duplicated to ensure each boundary neighbor contains the same number of points. We independently apply random multi-point boundary feature constraints to boundaries at different fineness degrees. The multi-fineness boundary feature constraint $\mathcal{L}_{MBC}$ can be formulated as follows:

$$\mathcal{L}_{MBC} = \frac{1}{M} \sum_{m=0}^{M-1} \lambda_m \sum_{x \in \mathcal{B}_m} \mathcal{L}_x. \quad (4)$$

Here, $\lambda_m$ represents the weight of the feature constraint for the *m-th* boundary query fineness degree. Boundary constraints at higher fineness degrees are assigned higher weights to ensure the accuracy of feature constraints. In contrast, those at lower fineness degrees are given lower weights to balance the completeness of boundary feature constraints. Finally, the total criterion of our MBSE framework is

$$\mathcal{L}_{MBSE} = \mathcal{L}_{seg} + \mathcal{L}_{MBC}. \quad (5)$$

Where $\mathcal{L}_{seg}$ stands for the loss of semantic segmentation. The above settings will be described in Section 4.1. The multi-fineness boundary constraints enhance the feature discrimination between boundaries and non-boundaries, reducing the difficulty of classification for the semantic segmentation discriminator. It improves the segmentation performance of boundary regions. The detailed evaluation and analysis of MBC is given in Section 4.4.1.

## 3.2 Shifted Ensemble-aware Perception

Introducing boundary feature constraints may distract the network from supervising segmentation, leading to problems such as unbalanced segmentation accuracy [43]. Long-range correlations enable the network to better comprehend the semantic structure of point clouds and discern differences between boundaries and non-boundaries across larger spatial extents. In this section, we introduce a new point cloud space called ensemble and propose a novel shifted ensemble-aware perception module. This module captures the direct point-to-point long-range correlations within ensembles. Additionally, It generalizes the ensemble patterns to the global point cloud with minimal computational cost, enhancing the overall segmentation performance.

**Figure 3: Diagram of the Ensemble-aware Perception Module, illustrating the process of obtaining ensemble-aware perception encoding within an ensemble. $F$ and $F'$ respectively represent the number of input features and the number of features after dimension reduction. $N$ represents the number of points in the ensemble.**

*3.2.1 **Ensemble Point Cloud Space.*** Long-range correlation refers to the association or relationship that exists between distant elements [25]. Capturing the long-range correlation can help the model identify semantic relationships between different parts of the point cloud, thereby improving semantic segmentation accuracy. Obtaining long-range correlations requires establishing remote information interaction. Current point cloud segmentation models [6, 29, 38] mainly rely on aggregating features in local neighbors, which consist of several tens of points, to obtain high semantic features for segmentation. These models' receptive fields gradually increase as the networks deepen and local features are aggregated. Obtaining relevant information between two non-adjacent points may require multiple rounds of feature abstractions or aggregations. In this complex process, some crucial long-range correlation information may be lost. However, due to the high volume of point cloud data, establishing direct global long-range interactions requires enormous computational costs.

To avoid this, we propose a novel concept in point cloud space called ensemble. The input point cloud $\mathcal{X}$ is randomly divided into $n$ non-overlapping ensembles $\epsilon_i$ (i.e., $\epsilon_i \subset \mathcal{X}$ and $\epsilon_i \cap \epsilon_j = \varnothing$). Each ensemble contains the same number of points. The spatial size of the ensemble depends on the size of the input point cloud, with the detailed configuration described in Section 4.1. Without extra computational overhead, the random partitioning of ensembles allows distant points in the point cloud to be grouped into the same feature encoding unit, facilitating direct information exchange between distant points. Direct information interactions and contextual relationships can be established within each ensemble, capturing long-range correlation. Compared to traditional local neighbor aggregation, ensembles extend the feature encoding space. The method for obtaining long-range correlation encodings within an ensemble is described in the following subsection.

*3.2.2 **Ensemble-aware Perception Encoding.*** The feature encodings embedded with long-range correlations are obtained in each ensemble. The diagram of the ensemble-aware perception module is illustrated in Figure 3. We take the inherent unordered nature of point clouds as prior knowledge, where semantic features should remain invariant to translation or rotation. Specifically, for each ensemble $\epsilon_i$, the difference between its position $p_{\epsilon_i}$ and the

average position of the point cloud $\overline{p_{\mathcal{X}}}$ is embedded into the input features $f_{\epsilon_i}$. This embedding ensures the position-invariant representation of the perception module and facilitates the generalization of the perception module across different ensembles in subsequent stages. This operation $\mathcal{F}$ can be formulated as follows:

$$\mathcal{F}(\epsilon_i) = h_{\theta_p}\left(\left[f_{\epsilon_i}, \frac{p_{\epsilon_i} - \overline{p_{\mathcal{X}}}}{n}\right]\right). \tag{6}$$

Where $[\cdot, \cdot]$ denotes the concatenation operation. $h_{\theta_p}$ is an operation implemented with an MLP that integrates features with position information and reduces the feature dimension to $1/s$ of its original size. $n$ denotes the number of ensembles in the point cloud. The output of $\mathcal{F}(\epsilon_i)$ serves as the initial encoding for the $i$-$th$ ensemble. The ensemble-aware perception module captures long-range correlations between points by computing associations and transforming the feature space. We argue that long-range correlations should exist in diverse aspects, such as semantics, geometries, color spaces, and density distributions. A single feature space transformation is insufficient to characterize point clouds' complex spatial and semantic features. We employ $G$ space transformations to capture long-range correlations within ensembles from multiple perspectives. Additionally, we perform contribution perception for different space transformations. The long-range correlation set $\mathcal{M}$, which encompasses $G$ long-range correlations for $\epsilon_i$, can be formulated as:

$$\mathcal{M}(\epsilon_i) = \left\{w_j \mathcal{A}^T(\epsilon_i)\mathcal{T}_j(\epsilon_i) \mid \forall j \in \{0, ..., G-1\}\right\}. \tag{7}$$

Here, $\mathcal{A}$ and $\mathcal{T}$ represent operations for feature alignment and space transformation, respectively, implemented by MLPs. $w_j$ denotes the contribution perception weight of the $j$-$th$ space transformation, implemented through a set of learnable parameters processed by Softmax function. Then, we multiply the $G$ long-range correlation matrixes with the initial feature encoding $\mathcal{F}(\epsilon_i)$ and sum them point-wise to integrate long-range correlation representations. The perception encoding of the $i$-$th$ ensemble can be formulated as:

$$\mathcal{H}(\epsilon_i) = h_{\theta_f}\left(O\left(\mathcal{F}(\epsilon_i)\mathcal{M}_j(\epsilon_i)\right)\right). \tag{8}$$

Here, $h_{\theta_f}$ represents an MLP used for feature alignment. $O$ stands for the operation of the point-wise matrix addition. $\mathcal{H}(\epsilon_i)$ contains ensemble-level long-range correlations. Without progressively

deepening the network and feature extraction, the ensemble-aware perception module's direct receptive field spans the ensemble rather than just the neighbor with several dozen points.

### 3.2.3 *Multi-layer Shifting and Parameter Sharing.* 

Hierarchical structures are widely applied in point cloud segmentation [6, 29, 38]. Existing models utilize hierarchical structures to downsample point clouds, obtaining multiple levels of sub-point clouds. With increasing downsamplings and feature abstraction iterations, high-layer sub-point clouds containing fewer points encapsulate richer semantic information conducive to semantic segmentation. While low-layer sub-point clouds with more points exhibit lower semantic feature levels, they compensate for detailed information such as position lost in high semantics. Ensemble-aware perception modules are applied separately at different layers to fully utilize features from different hierarchical layers. At lower layers, it enhances models' capability to capture spatial geometric patterns, while at higher layers, it strengthens global interactions among high-level semantic features. Like images, point clouds also possess positional invariance, necessitating ensemble-aware perception to be spatially agnostic. The ensemble-aware perception module shifts and shares its parameters across the point cloud to capture and generalize long-range correlations. We obtain the shifted ensemble-aware perception encoding of the point cloud $\mathcal{X}^l$ at the $l$-$th$ layer by concatenating its $n$ ensemble-aware perception encodings at spatial locations aligned with the input and establishing a shortcut connection, which can be formulated as follows:

$$\mathcal{K}^l(\mathcal{X}^l) = \left[ \{ \mathcal{H}^l(\epsilon_i) \mid \forall i \in \{0, ..., n-1\} \} \right] + f_{\mathcal{X}^l}. \quad (9)$$

Multi-layer shifting and parameter sharing establish an information exchange pathway from ensembles to the entire point cloud with low computational cost, further expanding the receptive fields. It enables the model to interact information with a global receptive field at different layers and generalize local patterns among different ensembles, thereby enhancing its ability to capture global spatial geometry and long-range correlations. The detailed evaluation and analysis of SEP is given in Section 4.4.2.

## 4 EXPERIMENTS

In this section, we showcase and analyze the performance of MBSE on point cloud semantic segmentation and part segmentation on multiple benchmarks. Through comparative experiments with various recent state-of-the-art methods and ablation studies, we demonstrate the effectiveness and superiority of our MBSE.

### 4.1 Implementation Details

We apply the MBSE framework to the popular and representative PointNext [38] family, taking its four variants as baselines. We add an SEP after each of their four sub-sampling modules and replace the loss function with our $\mathcal{L}_{MBSE}$. For fairness, we employ the same training and evaluation strategies as the baselines. Except for ShapePartNet [56], we do not use any voting strategy for evaluation. For MBC, we conduct boundary queries on three fineness degrees (i.e., $M = 3$). The boundary neighbor query distances $r_m$ and point numbers $k_m$ are set to (0.05, 0.1, 0.2) and (4, 8, 12), respectively. The number of random constraint key points is set to (2, 4, 6). The

constraint temperature $\tau$ is set to 1. The weights $\lambda_m$ for multi-fineness constraints are set to (0.5, 0.3, 0.2). For SEP, the downscale factor $s$ is set to 4. The four downsampled sub-point clouds are evenly divided into (36, 28, 20, 12) ensembles, respectively. The criterion used for segmentation (i.e., $\mathcal{L}_{seg}$) is CrossEntropy. 24k points are fed as a batch to train the models. We evaluate the models using the entire scene as input. Unless otherwise specified, all experiments are conducted according to the above settings.

### 4.2 3D Semantic Segmentation

For point cloud semantic segmentation, we evaluate our MBSE on the widely used S3DIS [1] and ScanNet [5]. S3DIS is a challenging large-scale 3D point cloud dataset. The results on S3DIS Area 5 are shown in Table 1. MBSE improves the performance of all Point-Next variants. Compared to the PointNext-XL [38], MBSE increases mIoU, OA, and mAcc by 1.3%, 0.4%, and 0.8%, respectively, and surpasses PointVector-XL [6] in terms of mIoU and OA. It achieves the best or second-best performance in IoU for almost half of all categories. Compared to PointVector, MBSE extends the direct receptive field to a larger ensemble level rather than the traditional neighbor. Based on the Transformer architecture, StratifiedFormer [25] has a larger receptive field, which is an advantage over the models based on MLPs. Nevertheless, MBSE performs highly competitively because it can capture direct correlations between points over long distances without requiring feature aggregation in neighbors. Furthermore, MBSE performs well in segmenting objects with high boundary proportions, such as beams and columns. On PointNext-L and PointNext-XL, the IoU of the beam is improved to 0.2% and 0.4% by MBSE, respectively, while this metric remains at 0.0% for almost all other comparative models.

We further evaluate MBSE on S3DIS Area 6-fold and ScanNet. The results are reported in Table 2. ScanNet comprises 1,613 3D indoor scans. We train and evaluate the models on training and validation sets, respectively. MBSE significantly improves the performance of all baselines. On S3DIS Area 6-fold, except for mIoU, the PointNext-XL model equipped with MBSE achieves the best performance. On ScanNet, the number of input points is set to 64k. The MBSE versions of PointNext-L and PointNext-XL achieve a 1.1% and 1.4% improvement in mIoU compared to the baselines, respectively. MBSE can provide more considerable performance boosts for larger baseline models. It is primarily due to larger models being able to extract richer high-level semantic information. Increasing feature discrimination and capturing long-range correlations for high semantic features yields superior semantic segmentation performance. To accurately evaluate the impact of MBSE on baselines, we did not modify the structure of the baselines. MBSE is applied with the same experimental settings on all baselines, with parameters and computational costs of 1M and 0.5 GFLOPs, respectively. The volume of the baselines primarily determines the parameters and computational costs of the models.

### 4.3 3D Object Part Segmentation

ShapeNetPart [56] is a widely used dataset for 3D object part segmentation. It contains a total of 16 different object categories with 16,880 models. Each category is comprised of 2 to 6 parts, resulting in a maximum of 50 labeled parts. The results are reported in Table

**Table 1: Comparison results on S3DIS Area 5. Consistent with PointNext, we use 24k input points, while the number of input points for StratifiedFormer is set to 80k. * denotes the method that also explicitly considers the boundary of the point cloud. The best and the second-best performance is bolded. The red indicates improvement over baseline.**

| Method | mIoU | OA | mAcc | ceiling | floor | wall | beam | column | window | door | table | chair | sofa | bookcase | board | clutter |
|---|---|---|---|---|---|---|---|---|---|---|---|---|---|---|---|---|
| PointNet [35] | 41.1 | - | 49.0 | 88.8 | 97.3 | 69.8 | 0.1 | 3.9 | 46.3 | 10.8 | 59.0 | 52.6 | 5.9 | 40.3 | 26.4 | 33.2 |
| PointCNN [28] | 57.3 | 85.9 | 63.9 | 92.3 | 98.2 | 79.4 | 0.0 | 17.6 | 22.8 | 62.1 | 74.4 | 80.6 | 31.7 | 66.7 | 62.1 | 56.7 |
| SPGraph [26] | 58.0 | 86.4 | 66.5 | 89.4 | 96.9 | 78.1 | 0.0 | 42.8 | 48.9 | 61.6 | 84.7 | 75.4 | 69.8 | 52.6 | 2.1 | 52.2 |
| KPConv [45] | 67.1 | - | 72.8 | 92.8 | 97.3 | 82.4 | 0.0 | 23.9 | 58.0 | 69.0 | 81.5 | 91.0 | 75.4 | 75.3 | 66.7 | 58.9 |
| MinkowskiNet [4] | 65.4 | - | 71.7 | 91.8 | **98.7** | **86.2** | 0.0 | 34.1 | 48.9 | 62.4 | 81.6 | 89.8 | 47.2 | 74.9 | 74.4 | 58.6 |
| JSENet* [23] | 67.7 | - | - | 93.8 | 97.0 | 83.0 | 0.0 | 23.2 | **61.3** | 71.6 | 89.9 | 79.8 | 75.6 | 72.3 | 72.7 | 60.4 |
| PCT [13] | 61.3 | - | 67.7 | 92.5 | 98.4 | 80.6 | 0.0 | 19.4 | 61.6 | 48.0 | 76.6 | 85.2 | 46.2 | 67.7 | 67.9 | 52.3 |
| PAConv [51] | 66.6 | - | 73.0 | 94.6 | 98.6 | 82.4 | 0.0 | 26.4 | 58.0 | 60.0 | 89.7 | 80.4 | 74.3 | 69.8 | 73.5 | 57.7 |
| Point Trans. [60] | 70.4 | 90.8 | 76.5 | 94.0 | 98.5 | **86.3** | 0.0 | 38.0 | **63.4** | 74.3 | 89.1 | 82.4 | 74.3 | **80.2** | 76.0 | 59.3 |
| StratifiedFormer [25] | 72.0 | **91.5** | **78.1** | 96.2 | **98.7** | 85.6 | 0.0 | **46.1** | 60.0 | **76.8** | 92.6 | 84.5 | 77.8 | 75.2 | **78.1** | 64.0 |
| CBL* [43] | 69.4 | 90.6 | 75.2 | 93.9 | 98.4 | 84.2 | 0.0 | 37.0 | 57.7 | 71.9 | **91.7** | 81.8 | 77.8 | 75.6 | 69.1 | 62.9 |
| RepSurf-U [40] | 68.9 | 90.2 | 76.0 | - | - | - | - | - | - | - | - | - | - | - | - | - |
| PointMetaBase [29] | 71.3 | 90.8 | - | - | - | - | - | - | - | - | - | - | - | - | - | - |
| PointVector-XL [6] | **72.3** | 91.0 | **78.1** | 95.1 | 98.6 | 85.1 | 0.0 | 41.4 | 60.8 | **76.7** | 84.4 | **92.1** | **82.0** | 77.2 | **85.1** | 61.4 |
| PointNext-S | 64.2 | 88.2 | 70.7 | 94.0 | 98.3 | 80.9 | 0.0 | 23.8 | 48.7 | 66.6 | 81.0 | 90.0 | 68.0 | 72.0 | 58.0 | 54.0 |
| +MBSE | 65.5 | 88.7 | 72.5 | 94.2 | 98.4 | 79.8 | 0.0 | 24.4 | 53.3 | 69.1 | 81.7 | 89.3 | 71.9 | 71.2 | 63.5 | 54.7 |
| PointNext-B | 67.5 | 89.4 | 73.9 | 93.8 | 98.4 | 82.3 | 0.0 | 18.9 | 55.8 | 74.3 | 82.2 | 90.8 | 76.5 | 74.8 | 71.8 | 58.0 |
| +MBSE | 68.4 | 89.7 | 74.3 | 93.4 | 98.3 | 82.8 | 0.0 | 27.6 | 57.0 | 72.9 | 83.2 | 91.4 | 75.7 | 74.0 | 77.9 | 55.2 |
| PointNext-L | 69.3 | 90.1 | 75.7 | 94.0 | 98.5 | 83.6 | 0.1 | 30.5 | 60.1 | 72.2 | 82.1 | 91.3 | 76.8 | 74.7 | 77.6 | 59.9 |
| +MBSE | 70.1 | 90.6 | 76.3 | 93.8 | **98.7** | 84.0 | **0.2** | 37.1 | 59.6 | 73.9 | 87.2 | 89.7 | 78.3 | 75.6 | 73.9 | 58.7 |
| PointNext-XL | 71.1 | 91.0 | 77.2 | 93.7 | 98.6 | 85.3 | 0.0 | 42.3 | 60.6 | 70.9 | 84.4 | **92.4** | 80.4 | 78.1 | 76.6 | 61.0 |
| +MBSE | **72.4** | **91.4** | **78.0** | 95.4 | **98.9** | 85.9 | **0.4** | 45.4 | 60.9 | 75.9 | 86.8 | 91.9 | 79.7 | **79.1** | 77.9 | 62.5 |

**Table 2: Comparison results on Area 6-fold and ScanNet.**

| Method | S3DIS Area 6-fold | | | ScanNet | Params | FLOPs |
|---|---|---|---|---|---|---|
| | mIoU | OA | mAcc | mIoU | M | G |
| PointNet [35] | 47.6 | 78.5 | 66.2 | - | 3.6 | 35.5 |
| DGCNN [48] | 56.1 | 84.1 | - | - | 1.3 | - |
| DeepGCN [27] | 60.0 | 85.9 | - | - | 3.6 | - |
| KPConv [45] | 70.6 | - | 79.1 | 69.2 | 15.0 | - |
| RandLA-Net [21] | 70.0 | 88.0 | 82.0 | - | 1.3 | 5.8 |
| MinkowskiNet [4] | - | - | - | 72.2 | 37.9 | - |
| PointASNL [53] | 68.7 | 88.8 | 79.0 | 63.5 | 22.4 | 19.1 |
| BAAF [39] | 72.2 | 88.9 | 83.1 | - | 5.0 | - |
| Point Trans. [60] | 73.5 | 90.2 | 81.9 | 70.6 | 7.8 | 5.6 |
| CBL [43] | 73.1 | 89.6 | 79.4 | - | 18.6 | - |
| PointMetaBase [29] | 77.0 | 91.3 | - | 72.8 | 19.7 | 11.0 |
| PointVector [6] | **78.4** | 91.9 | 86.1 | - | 24.1 | 58.5 |
| PointNext-S | 68.0 | 87.4 | 77.3 | 64.5 | 0.8 | 3.6 |
| +MBSE | 70.3 | 88.6 | 78.7 | 65.2 | 1.8 | 4.1 |
| PointNext-B | 71.5 | 88.8 | 80.2 | 68.4 | 3.8 | 8.9 |
| +MBSE | 72.9 | 89.7 | 81.5 | 69.6 | 4.8 | 9.4 |
| PointNext-L | 73.9 | 89.8 | 82.2 | 69.4 | 7.1 | 15.2 |
| +MBSE | 75.7 | 91.2 | 84.7 | 70.5 | 8.1 | 15.7 |
| PointNext-XL | 74.9 | 90.3 | 83.0 | 71.5 | 41.6 | 84.8 |
| +MBSE | 77.8 | **92.2** | **86.4** | **72.9** | 42.6 | 85.3 |

3. To ensure fair comparison experiments, following PointNext, we employed voting by averaging the results of 10 randomly scaled input point clouds, with scaling factors ranging from 0.8 to 1.2. The input point number is set to 2,048. We conduct boundary queries on two fineness degrees (i.e., $M = 2$), with query distances $r_m$ of 0.05 and 0.1 and query numbers $k_m$ of 4 and 8, respectively. The number of random constraint key points is set to 2 and 4. The weights of multi-fineness constraint $\lambda_m$ are set to 0.7 and 0.3. Focal-Loss is used as the segmentation criterion. The input point cloud is downsampled into four sub-point clouds, each containing (8, 4, 2, 1) ensembles, respectively. The MBSE version of PointNext achieved the best performance in both instance mIoU and class mIoU.

## 4.4 Ablation and Analysis

To further verify the effectiveness of our MBSE, we perform ablation studies on S3DIS Area 5 with PointNext-XL as the baseline.

*4.4.1 The Effectiveness of MBC.* We conduct detailed comparisons of the different boundary query methods, query fineness degrees, and feature constraint methods in MBC. Additionally, we show the performance of the beam and the column with high boundary proportions. From Table 4, it can be observed that the boundary query method affects the semantic segmentation accuracy. The segmentation accuracy of the multi-boundary version with KNN is even lower than that of the single-boundary version, as looser boundary queries on KNN may result in more erroneous boundaries. Ball queries limit the range of neighbor queries, improving the quality of boundary queries. In the multi-boundary setting, ball queries improve 0.6%, 1.0%, and 1.7% in mIoU, OA, and mAcc, respectively. Multi-fineness boundaries with ball queries substantially

**Table 3: Comparison results on ShapeNetPart.**

| Method | ins. mIoU | cls. mIoU |
|---|---|---|
| PointNet [35] | 83.7 | 80.4 |
| SpiderCNN [52] | 85.3 | 82.4 |
| RS-CNN [30] | 86.2 | 84.0 |
| KPConv [45] | 86.4 | 85.1 |
| CurveNet [50] | 86.8 | - |
| ASSANet [37] | 86.1 | - |
| Point Transformer [60] | 86.6 | 83.7 |
| PointMLP [32] | 86.1 | 84.6 |
| StratifiedFormer [25] | 86.6 | 85.1 |
| PointVector [6] | 86.9 | - |
| PointMetaBase [29] | 87.1 | 85.1 |
| PointNext-S | 87.0 | 85.2 |
| **+MBSE** | **87.6** | **85.5** |

enhance the performance of objects with high boundary proportions, increasing the IoU of beams and columns by 0.2% and 1.7% over single-boundary, respectively, demonstrating the effectiveness of multi-fineness boundary queries. The random multi-point constraint effectively enhances the feature discrimination quality between boundaries and non-boundaries compared to single center point constraints. MBC notably improved the segmentation performance of objects with high boundary proportions. However, the overall performance improvement is modest, mainly because introducing boundary constraints may divert the model's attention from supervising the segmentation.

**Table 4: The results of different settings on the MBC. Except for the changes stated in each row, the rest of the settings are set to the defaults to control variables.**

| Method | mIoU | OA | mAcc | beam | column |
|---|---|---|---|---|---|
| baseline | 71.1 | 91.0 | 77.2 | 0.0 | 42.3 |
| single bound. (knn) | 70.9 | 90.7 | 77.1 | 0.0 | 41.0 |
| multi-bounds. (knn) | 70.8 | 90.1 | 75.7 | 0.0 | 38.9 |
| single bound. (bq.) | 71.2 | 91.0 | 77.0 | 0.0 | 42.5 |
| single constraint | 71.2 | 90.8 | 77.3 | 0.1 | 43.7 |
| **MBC (default)** | **71.4** | **91.1** | **77.4** | **0.2** | **44.2** |

*4.4.2* ***The Effectiveness of SEP****.* To explore the effectiveness of SEP, we conducted ablation studies on its key components. The results of its performance and computational cost are reported in Table 5. Removing the relative position embedding led to a decrease in all metrics, demonstrating the effectiveness of position-invariant representation. Non-shared parameter results in a 4.8M increase in the number of parameters and a sharp drop in performance. The main reason is that ensembles are randomly sampled sub-point clouds and the non-shared parameter poses challenges for pattern convergence. Furthermore, we explore how the number of space transformation blocks affects SEP. Increasing the number of blocks enables the model to perceive the interrelations between points

from more aspects, capturing long-range correlations comprehensively. However, an excessive number of the blocks might overfit the training data, leading to a decrease in performance. After comparison, we set the default number of space transformation blocks to 8. Different long-range correlations contribute differently to the segmentation. Contribution perception enabled SEP to achieve better performance. SEP boosted the baseline's mIoU, OA, and mAcc by 0.9%, 0.2%, and 0.6% in low computational cost, proving the effectiveness of SEP.

**Table 5: The results of different settings on the performance and computational cost in SEP. The values in Param and GFLOPs represent the variations relative to the complete SEP with default settings. ~0 indicates a negligible change in computational cost, which is difficult to reflect in the evaluation metrics due to orders of magnitude.**

| Method | mIoU | OA | mAcc | Param | GFLOPs |
|---|---|---|---|---|---|
| baseline | 71.1 | 91.0 | 77.2 | -1.0 | -0.5 |
| w/o pos. embedding | 71.8 | 91.0 | 77.4 | ~ 0 | ~ 0 |
| non-shared parameter | 69.6 | 88.7 | 74.9 | +4.8 | 0 |
| one space transformation | 71.3 | 91.1 | 77.2 | -0.6 | -0.3 |
| 16 space transformations | 70.7 | 89.5 | 76.8 | 1.3 | 0.7 |
| w/o contribution percep. | 71.7 | 91.1 | 77.4 | ~0 | ~0 |
| **SEP (default)** | **72.0** | **91.2** | **77.8** | - | - |

*4.4.3* ***Relationship between MBC and SEP****.* We analyze the relationship between MBC and SEP. Both MBC and SEP stably improve the baseline performance. MBC demonstrated a relatively modest improvement compared to SEP, as its emphasis lies primarily on enhancing boundary segmentation. Compared to solely applying either MBC or SEP, MBSE further improves the semantic segmentation performance of the model. This is because SEP further generalizes and optimizes the feature discrimination between boundaries and non-boundaries constrained by MBC across the global point cloud. The complementarity of the two enables MBSE to achieve excellent segmentation performance on both boundaries and the overall point cloud.

## 5 CONCLUSION

We conduct the first systematic study on the impact of boundary queries on the semantic segmentation of point clouds. We introduce a multi-fineness boundary constraint approach, which explicitly enhances the feature discrimination between boundaries and non-boundaries, improving the accuracy of difficult-to-segment categories with high boundary proportions. Additionally, we proposed a shifted ensemble-aware perception module that establishes direct point-to-point interactions in novel ensemble spaces. With low computational costs, SEP effectively generalizes the discrimination patterns between boundaries and non-boundaries and the long-range correlations across the entire point cloud. Experimental results demonstrate that MBSE significantly improves segmentation performance for classes with high boundary proportions and the overall point cloud. We hope this work will inspire further exploration into studying the explicit enhancement of boundary segmentation and the long-range correlations in point clouds.

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
