# OpenReview forum: "Multi-fineness Boundaries and the Shifted Ensemble-aware Encoding for Point Cloud Semantic Segmentation"
_acmmm.org/ACMMM/2024/Conference — MM2024 Poster_

### Official Review · Reviewer_nySc · 2024-05-09

**Rating:** 4
**Confidence:** 3

**Summary:**

This paper attempt to deal with 3D point cloud segmentation. It proposed Multi-fineness Boundary Constraint to enhance the feature discrimination in boundary areas. It also proposed shifted ensemble-aware perception to learn the long-range relationship. Results in S3DIS Area 5, ScanNet, and ShapeNetPart shows improvement over backbone.

**Strengths:**

1. The motivation make sense to me to utilize contrastive loss in boundary area and learn the long-range relationship.
2. Experiments on various datasets shows the improvement over backbone PointNext.

**Limitations:**

1. The definition of ensemble is not clear, whether it is similar to super-point or it is formed by collecting points randomly from a point cloud. There is no visual illustration. Points are grouped according to the feature in “Dynamic Graph CNN for Learning on Point Clouds”, is ensemble a better way to learn the long-range relationship?
2. The notation like A and H in Eq 7 and Eq 8 is not shown in the Figure 3 making it hard to understand the design of SEP.
3. The constrain in the boundary area is similar to the region similarity loss in “SceneEncoder: Scene-Aware Semantic Segmentation of Point Clouds with A Learnable Scene Descriptor”, whether such constrain works better.
4. Only PointNext is utilized as the backbone, it will be better to show the improvement on other backbone like “Kpconv: Flexible and deformable convolution for point clouds”or “4d spatio-temporal convnets: Minkowski convolutional neural networks” to prove the generalization ability.

**Suitability:**

2

---

### Official Review · Reviewer_d74P · 2024-05-16

**Rating:** 4
**Confidence:** 1

**Summary:**

This paper introduces a method called Multi-fineness Boundary Constraint (MBC) to tackle the challenge that the boundaries of point cloud are prone to mis-segmentation. By querying boundaries at various degrees of fineness and imposing feature constraints within these boundary areas, the discrimination between boundaries and non-boundaries is enhanced. Moreover, a new concept of point cloud space termed ensemble and a Shifted Ensemble-aware Perception (SEP) module is introduced to mitigate the problem that solely emphasizing boundaries may compromise the segmentation accuracy in broader non boundary regions. This module establishes information interactions between points with minimal computational cost, effectively capturing direct point-to-point long-range correlations within ensembles. The experimental results on multiple benchmarks validate the effectiveness and superiority of proposed method. However, considering the compared baselines are relatively weak and I am not familiar with the area, I recommend borderline accept score.

**Strengths:**

1.This paper conducts the first systematic study on the impact of boundary queries on the semantic segmentation of point clouds. It introduces a multi-fineness boundary constraint approach, which explicitly enhances the feature discrimination between boundaries and non-boundaries, improving the accuracy of difficult-to-segment categories with high boundary proportions.

2.The experimental results demonstrate the effectiveness of the proposed method, including Multi-fineness Boundary Constraint (MBC) and Shifted Ensemble-aware Perception (SEP).

3.The paper is well-written with all key concepts are well described and easy to follow.

**Limitations:**

1.The Comparative experiments are not convincing. Several state-of-the-art papers are ignored, such as PTv2 [1], Cluster3DSeg [2], PTv3 [3] and so on. Please compare with these strong baselines.

2.The experiments for adaptable of proposed method are not enough. Only PointNext are adopted to present results with proposed MBSE. It is better to combine with more strong baselines to make the adaptable ability more convincing.

[1]Point transformer v2: Grouped vector attention and partition-based pooling, NeurIPS 2022.

[2]Clustering based Point Cloud Representation Learning for 3D Analysis, ICCV 2023.

[3]Point Transformer V3: Simpler, Faster, Stronger, CVPR 2024.

**Suitability:**

2

---

### Official Review · Reviewer_M3h5 · 2024-05-20

**Rating:** 4
**Confidence:** 3

**Summary:**

This paper introduces a Multi-finenessBoundary Constraint (MBC) method to improve point cloud boundaries for semantic segmentation. By querying boundaries of different fineness and imposing feature constraints in these boundary areas, the distinction between boundaries and non-boundaries is enhanced. In addition, Shifted Ensemble-aware Perception (SEP) module is proposed, which establishes information interaction between points with minimal computational cost, captures the direct point-to-point long-range correlation in the ensemble and improves the segmentation performance of boundaries and non-boundaries. Experiments demonstrate the effectiveness of the proposed method.

**Strengths:**

1.The proposed MBC can enhance the feature distinction between boundaries and non-boundaries and improve segmentation decisions without additional networks.
2.The proposed SEP module captures direct point-to-point long-range dependencies in a collection while keeping the computational requirements low.
2.This manuscript designs many experiments to demonstrate the effectiveness of the proposed methods.

**Limitations:**

1. In section 3.2.2, the manuscript states that “We take the inherent unordered nature of point clouds as prior knowledge”. Since the point cloud is unordered, how is the location information of the point cloud expressed?
2. Please provide more details about feature alignment, and what operations can achieve feature alignment?
3. EPM captures the long-term correlation between points by multiplying matrices. However, the result of the matrix product calculation represents the similarity of the two matrices, and it does not seem to be effectively related to long-range correlations simply through the matrix product operation.
4. Many hyperparameters are used in the proposed methods. However, the experimental section lacks analysis on hyperparameters, such as r, k or the number of ensembles. Do these hyperparameters affect algorithm stability, and how to carefully design a reasonable range?

**Suitability:**

2

---

### Official Review · Reviewer_HmgC · 2024-05-27

**Rating:** 2
**Confidence:** 3

**Summary:**

The author introduces Multi-fineness Boundary Constraint (MBC) and shifted ensemble-aware perception (SEP). The former enhances the discrimination between boundaries and non-boundaries, thereby improving point cloud boundary segmentation. The latter delicately captures direct point-to-point long-range correlations within ensembles. Extensive experiments validate the effectiveness of the proposed method.

**Strengths:**

Without additional networks, it enhances the feature discrimination between boundaries and nonboundaries, assisting the discriminator in making better segmentation decisions for boundaries.

**Limitations:**

Weakness:
1. The motivation of this paper is to emphasize the importance of boundary query for segmentation prediction, but there is no “BIoU” to measure the effectiveness of the proposed MBC in the experimental results. Please explain the reason and make experimental supplements.
2. Line 346-350, this part is confused, the author states that the current method based on boundary query only relies on K nearest neighbors and lacks distance restrictions, but in CBL, the boundary query is constrained by a constant radius of 0.1. Can an identical distance constraint (0.1) be given to measure the effectiveness of “Multi-point Boundary Feature Constraints”?
3. For query distances r_m and point numbers k_m, whether there is any basis for comparison of results, e.g. “fixed r_m and adjusted k_m” or “fixed k_m and adjusted r_m”.
4. In Eq.(3), what does the meaning of notation f_p, f_j, f_v?
5. In line 501, the author divides the input into n non-overlapping ensembles. I do not see the detailed configuration of splits in section 4.1.
6. The method proposed by the author is similar to CBL in both its core idea and method framework. Please supplement the comparison of the results with CBL under the same network. (e.g. PointNext-B + CBL or RandLA-Net + MBSE)
7. In summary, lack of novelty. MBC looks more like a trick in terms of experimental results and method details. There is no effective basis for selecting Random multiple points. Compared to single selection, multi-point or multi-prototype is always the better choice.
8. As I understand, ConDaFormer [Ref1] can achieve 73.5% mIOU on S3DIS Area 5. What is the competitiveness of your method?
[Ref1] Duan, Lunhao, et al. "ConDaFormer: Disassembled Transformer with Local Structure Enhancement for 3D Point Cloud Understanding." Advances in Neural Information Processing Systems (2024).

Suggestion:
1. “enclave phenomenon” is a confusing word without citing the source, and if the definition proposed by the author himself needs to be explained or proved.
2. Losing visualization results with competitors.

**Suitability:**

2

---

### Meta-Review · Area_Chair_ECu9 · 2024-06-28

**Recommendation:** Accept (Poster)
**Confidence:** 4

**Metareview:**

Initially the paper received 3 positive and 1 negative ratings. After rebuttal, one reviewer raised the rating, and thus all four reviewers reached a consensus on "borderline accept". After carefully reading both the reviews and the paper, the AC agrees with the reviewers' recommendations, and concludes that the submission hold values in proposing a novel method to enhance semantic segmentation of point cloud boundaries. However, the paper also has some weaknesses such as lack of testing with more recent backbones, marginal improvements, etc. The AC leans towards an acceptance, and would encourage the authors to further improve the quality of camera ready paper by carefully referring to the reviewers comments.

---

### Meta-Review · Senior_Area_Chairs · 2024-07-10

**Recommendation:** Accept (Poster)
**Confidence:** 4

**Metareview:**

This paper received mixed ratings initially. After rebuttal, all the reviewers tend to accept the paper. SAC and AC agree with reviewers and recommend acceptance of the paper.